# Effervescent Atomizer as Novel Cell Spray Technology to Decrease the Gas-to-Liquid Ratio

**DOI:** 10.3390/pharmaceutics14112421

**Published:** 2022-11-09

**Authors:** Anja Lena Thiebes, Sarah Klein, Jonas Zingsheim, Georg H. Möller, Stefanie Gürzing, Manuel A. Reddemann, Mehdi Behbahani, Stefan Jockenhoevel, Christian G. Cornelissen

**Affiliations:** 1Department of Biohybrid & Medical Textiles (BioTex), AME—Institute of Applied Medical Engineering, Helmholtz Institute Aachen, RWTH Aachen University, Forckenbeckstraße 55, 52074 Aachen, Germany; 2Aachen-Maastricht Institute for Biobased Materials, Faculty of Science and Engineering, Maastricht University, Brightlands Chemelot Campus, Urmonderbaan 22, 6167 RD Geleen, The Netherlands; 3Laboratory Biomaterial, University of Applied Sciences Aachen, Heinrich-Mußmann-Straße 1, 52428 Jülich, Germany; 4Institute of Heat and Mass Transfer (WSA), RWTH Aachen University, Augustinerbach 6, 52056 Aachen, Germany; 5Department of Pneumology and Internal Intensive Care Medicine, Medical Clinic V, University Hospital RWTH Aachen, Pauwelsstraße 30, 52074 Aachen, Germany

**Keywords:** cell aerosolization, cell atomization, adipose-derived stromal cells (ASCs), twin-fluid atomizer, survival, tri-lineage differentiation

## Abstract

Cell spraying has become a feasible application method for cell therapy and tissue engineering approaches. Different devices have been used with varying success. Often, twin-fluid atomizers are used, which require a high gas velocity for optimal aerosolization characteristics. To decrease the amount and velocity of required air, a custom-made atomizer was designed based on the effervescent principle. Different designs were evaluated regarding spray characteristics and their influence on human adipose-derived mesenchymal stromal cells. The arithmetic mean diameters of the droplets were 15.4–33.5 µm with decreasing diameters for increasing gas-to-liquid ratios. The survival rate was >90% of the control for the lowest gas-to-liquid ratio. For higher ratios, cell survival decreased to approximately 50%. Further experiments were performed with the design, which had shown the highest survival rates. After seven days, no significant differences in metabolic activity were observed. The apoptosis rates were not influenced by aerosolization, while high gas-to-liquid ratios caused increased necrosis levels. Tri-lineage differentiation potential into adipocytes, chondrocytes, and osteoblasts was not negatively influenced by aerosolization. Thus, the effervescent aerosolization principle was proven suitable for cell applications requiring reduced amounts of supplied air. This is the first time an effervescent atomizer was used for cell processing.

## 1. Introduction

In tissue engineering and cell therapy, methods to apply cells in various manners are needed. Thus, the interest in cell spraying (aerosolization) has increased over the last few years. First described by Bahoric et al. in 1997 for the application of epidermal cells [1], several studies have been published since with different spraying principles, cells, and applications with varying success [2,3,4,5,6,7]. The application of cells via sprays is particularly interesting to (i) coat uneven surfaces with a layer of cells, and (ii) deliver cell suspensions endoscopically in situ. For the latter, lung cell therapy can be mentioned as an interesting application to spray cells intrapulmonary, as it allows a local delivery and distribution of the cells within the diseased tissue [8,9]. For (i), a prominent example is the application of keratinocytes to chronic or burn wounds [2,5]. Here, spraying allows an even and widespread distribution of cells (in combination with a glue or hydrogel) to apply and keep the cells at the injured site. Most commonly used are twin-fluid aerosolization devices, which utilize pressurized gas to form droplets (airbrush, coaxial atomizer).

Additionally, the authors as well as other groups have developed and tested devices for intrapulmonary cell spraying for cell therapy approaches for lung diseases [8,9,10]. In a recent study, the authors compared a coaxial atomizer with a pressure based-aerosolization device, in which droplet formation is caused by the pressure drop at the exit of the nozzle [8]. While survival and metabolic activity remained unchanged, higher toxicity rates were detected for the pressure-based device. Thus, spraying with a coaxial atomizer is more suitable for cell application. Nevertheless, there is still room for improvement as the high air volumes in coaxial atomizers might lead to (i) quick evaporation, and (ii) draining of the sprayed liquid. Thus, it might be beneficial to reduce the amount of air in air-based atomizers. Previously, it was observed that a lower gas-to-liquid ratio could result in higher cell survival [11]. For cell therapy approaches within the body, less air injection would be advantageous, too, to avoid massive inflation of the organ in which the cell suspension is applied by the air needed for spraying.

Thus, this study focuses on a special twin-fluid atomization principle that allows the reduction of applied air with a specific setup: the effervescent atomizer. For conventional twin-fluid atomizers, air and liquid mix at the outlet of the liquid tube (Figure 1a), whereas in an effervescent atomizer, the air is bubbled into the liquid via holes within the liquid tube (Figure 1b), thereby reducing the needed ratio of gas to liquid. In addition, effervescent atomizers have the advantages that only a low air velocity is needed, exit orifices can be bigger, a variety of liquid viscosities can be aerosolized, and flow velocities at the outlet are generally lower [12]. Effervescent atomizers are used for consumer spray products, gas turbine combustors, or internal combustion engines [12].

In this study, a custom-made effervescent atomizer was designed to be tested with cell application. Three different mixing tubes, varying in number and diameter of holes for air bubbling of the liquid, were evaluated for four different gas-to-liquid ratios concerning the arithmetic mean diameter (D_10_) and droplet size distributions. Subsequently, human adipose-derived stromal cells (ASCs, often referred to as adipose-derived mesenchymal stromal cells, adMSCs) were aerosolized and the cell survival, as well as the cell behavior, were assessed, the latter by means of metabolic activity, apoptosis, and necrosis, as well as the expression of cell-specific surface marker and tri-lineage differentiation potential. To the authors’ knowledge, this is the first time an effervescent atomizer is used for cell processing.

## 2. Materials and Methods

### 2.1. Design of Custom-Made Effervescent Atomizers

To investigate the atomization characteristics and sprayed cells, a custom-made effervescent atomizer was designed with three different mixing tubes (Figure 2). The mixing tube in combination with the nozzle tip and the housing forms the atomizer (Figure 3). For these, the number, orientation, and diameter of the holes were varied as given in Table 1.

For cell spraying, all parts in contact with cell suspension were made of biocompatible materials that support steam sterilization. Housing, mixing tubes, and nozzle tip are made of polyether-ether-ketone (PEEK) and all seals are made of polydimethylsiloxane (PDMS) or polytetrafluorethylene (PTFE).

For a possible implementation of the effervescent atomizer in intrapulmonary cell spraying applications, the geometrical dimensions of this atomizer are small in comparison to conventional effervescent atomizers used for industrial combustion [13]. Therefore, the size of the inner diameter of the mixing tube is chosen to be similar to the inner diameter of previously developed coaxial atomizers [14], in order to simplify a subsequent development of a prototype for use in bronchoscopes. The further internal geometry is designed according to Jedelsky et al. [15], who measured the dependence of the Sauter mean diameter of the spray on the geometrical conditions of several atomizers. The SMD describes the diameter with the same volume-to-surface ratio as the spray. According to Jedelsky et al., the variation in the number of drilling holes has no clear influence on the SMD, but the increase in the number of drilling circles and the total area of holes tendentially decreases the SMD. Thus, the total area of holes defines the velocity of the entering air stream, and the influence of the diameter change on the SMD is expected to be low. Therefore, the lowest SMD and hence the lowest arithmetic mean diameter (D_10_) is expected for mixing tube C with the highest total area of holes and the highest number of drilling circles.

### 2.2. Spray Setup and Test Parameters

The experimental setup for spraying with the custom-made effervescent atomizer is shown in Figure 3. A syringe (B. Braun Melsungen) is mounted on a syringe pump (LA-120, Landgraf HLL) and connected to the effervescent atomizer via disposable tubing (extension line type Heidelberger, Fresenius Kabi). Compressed air is connected to the housing of the atomizer with two sterile gas filters on opposite sides of the housing. A rotameter (FLDA3420C, Omega) and pressure gauge are used to adjust gas flow. For sterile conditions during cell spraying, the atomizer was set up under a laminar flow hood.

The investigated spray parameters are shown in Table 2. For each mixing tube, four different boundary conditions with increasing gas-to-liquid ratios between 2.5% and 10% were tested. For all conditions, the flow rate of the spray fluid was set to 24 mL/min. The excess air pressure was increased to 2 bar for the gas-to-liquid ratio 10 to be able to keep the flow rate constant.

### 2.3. Spray Visualization and Evaluation of Droplet Size Distribution

The spray was visualized using a custom-made high-speed microscopic imaging system as previously described [11]. In brief, a long-distance transmitted light microscopy system [16] is combined with a monochromatic and incoherent pulsed light source (Cavilux Smart, 640 nm wavelength, 10 ns pulse duration and a high-speed camera (Fastcam SA-X, Photron, 25,000 frames per second) to capture high-speed videos of the primary spray break-up in the vicinity of the nozzle orifice and the droplet propagation in an adequate distance from the nozzle, ensuring steady-state atomization conditions. For capturing the whole spray width, each condition is measured by three videos with a duration of 200 ms (5000 frames) at different radial positions of the spray axis. The resolution is set to 1.614 µm/pixel.

To determine droplet sizes, the high-speed videos were analyzed frame by frame by converting each frame into a binary image and characterizing droplets by a custom-made Matlab (MathWorks) script within a range of 6 and 500 µm diameter. An inter-framing time of 0.4 ms (10 frames) ensures that droplets are not detected multiple times in successive frames. The data was used to compute the share of droplets (constant bin size of 2 µm) as well as the arithmetic mean diameter (D_10_) of the droplets. D_10_ is defined as the average of the diameters of all droplets in the spray sample.

### 2.4. Isolation and Culture of Adipose-Derived Stromal Cells

Human ASCs were isolated from subcutaneous fat tissue according to established protocols [17]. Fat tissue was kindly provided by the Clinic for Oral and Maxillofacial Surgery of the RWTH Aachen University Hospital following approval by the local ethics committee of the Medical Faculty of RWTH Aachen University (EK 067/18) and informed written consent provided by the patients. Briefly, minced fat tissue was enzymatically dissociated by incubation with 1 mg/mL collagenase (Sigma-Aldrich, Darmstadt, Germany) in a gentleMACS dissociator (Miltenyi Biotec, Bergisch Gladbach, Germany). Isolated ASCs were cultured in tissue culture flasks with Mesenpan culture medium containing growth supplements (both PAN-Biotech), 2% fetal calf serum (FCS), and 1% antibiotic-antimycotic solution (ABM, both ThermoFisher Scientific, Darmstadt, Germany) in a humidified incubator at 37 °C and 5% CO_2_. Cells in passages 2 and 3 sourced from three independent donors (*n* = 3) were used for all experiments.

### 2.5. Spraying of Adipose-Derived Stromal Cells

After trypsination, ASCs were suspended in phosphate-buffered saline (PBS, Gibco) at a concentration of 0.5 × 10^6^ cells/mL, stored at room temperature (RT) until use, and sprayed within 30 min of preparation. The cell suspension was transferred to a syringe (B. Braun Melsungen) and mounted on the syringe pump. The spraying distance from nozzle tip to substrate (polypropylene beaker) was set to 10 cm. The airflow was started and set to the test parameters. Once a steady state was reached, at least 2 mL of cell suspension was sprayed and the sample was collected. Until further processing or analysis, cells were stored on ice in the dark. In between spraying of different donors, the system was rinsed by spraying with sterile PBS.

### 2.6. Evaluation of Cell Survival via Flow Cytometry

For subsequent evaluation of cell survival of sprayed cells, cells were initially stained with the cell-permeable dye Calcein AM (AAT Bioquest) before spraying. In viable cells, calcein AM is converted to a green-fluorescent calcein after acetoxymethyl ester hydrolysis by intracellular esterases. With a concentration of 1.0 × 10^6^ cells/mL, ASCs were suspended in 4 µM Calcein AM in PBS and incubated, protected from light for 20 min at 37 °C and 5% CO_2_. The cell suspension was centrifuged at 500× *g* for 5 min and re-suspended in PBS at a concentration of 0.5 × 10^6^ cells/mL to prepare the final spray solution.

After spraying, cell suspensions were homogenized and diluted with Dulbecco’s Modified Eagle’s Medium (DMEM) plus 10% FCS at a ratio of 1:1 and stored, protected from light on ice. Non-sprayed cells served as controls. Before analysis by flow cytometry, samples were gently vortexed and filtered with a strainer (pore size 70 µm). Flow cytometry was performed with a FACSCanto II and BD FACSDiva Software 8.0.3 (both Becton Dickinson, Heidelberg, Germany). Cells were analyzed within 4 h after spraying. Events were first gated to exclude doublets and debris by means of the forward scatter-sideward scatter plot (FSC/SSC-plot), then gated to show events for calcein-positive and -negative populations by means of a histogram showing the event count in dependence of the fluorescent intensity. All gates were set globally.

Based on the higher survival rates for mixing tube B, all following evaluations and assays were carried out for cells sprayed with mixing tube B. For the assays, all four gas-to-liquid ratios were tested. For evaluation of tri-lineage potential and expression of characteristic mesenchymal stromal cell surface markers, the gas-to-liquid ratios 2.5% and 10% were analyzed.

### 2.7. Evaluation of Cell Behavior

To evaluate the influence of spraying on cell health, assays for metabolic activity (Cell Proliferation Kit II (XTT), Roche, Mannheim, Germany), apoptosis (Caspase-Glo 3/7 assay, Promega, Walldorf, Germany), and necrosis (ToxiLight BioAssay, Lonza, Cologne, Germany) were performed according to the manufacturers’ instructions. For this, cells (*n* = 3) were sprayed at a concentration of 0.5 × 10^6^ cells/mL PBS, diluted with full Mesenpan medium, and seeded in 96-well plates at a concentration of 1.6 × 10^3^ cells/cm^2^.

#### 2.7.1. Apoptosis

The Caspase Glo 3/7 assay is a luminescent assay to measure caspase-3/7 activity as a measure for the induction of apoptosis. Non-sprayed cells and cells incubated for 6 h with 1 µM staurosporine (Alfa Aesar, Kandel, Germany) served as controls. Apoptosis was evaluated after 24 h. Caspase-Glo 3/7 reagent was added to each well and the cells were incubated for 30 min at RT. Luminescence was measured by a spectrophotometer (Infinite M200, Tecan, Crailsheim, Germany) with an integration time of 100 ms.

#### 2.7.2. Necrosis

The ToxiLight assay evaluates the necrosis based on the release of adenylate kinase from damaged cells. The enzyme phosphorylates ADP and the resultant ATP is then measured using the bioluminescent firefly luciferase reaction. The assay was performed 24 h after cell spraying. Non-sprayed cells and cells lysed with ToxiLight 100% Lysis Control Set (Lonza) served as controls. In brief, adenylate kinase reagent was transferred to each well, and samples were incubated for 5 min at RT. Luminescence was measured by a spectrophotometer (Infinite M200, Tecan) with an integration time of 1000 ms.

#### 2.7.3. Metabolic Activity

The XTT assay is a colorimetric assay that detects cellular metabolic activities by reduction of the yellow tetrazolium salt (2,3-bis(2-methoxy-4-nitro-5-sulfophenyl)-2H-tetrazolium-5-carboxanilide, XTT) to a colored formazan. This conversion occurs only in living cells, therefore the amount of formazan produced is proportional to the number of living cells in the sample. For the evaluation, sprayed and non-sprayed cells were allowed to adhere for 24 h before the medium was changed to the respective testing medium. As suggested by ISO 10993, a full Mesenpan medium previously incubated for 72 h at 37 °C and 5% CO_2_ with toxic latex served as control. For uniform test conditions, all media in contact with cells were previously incubated for 72 h at 37 °C and 5% CO_2_ and stored at 4 °C until use. All media were changed daily. XTT was performed 48, 96, and 144 h after spraying. In brief, electron coupling reagent and XTT solution were freshly mixed at a ratio of 1:50 and 50 µL of working solution were transferred to each well. Cells were incubated for 4 h at 37 °C and 5% CO_2_ and conversion of XTT to formazan was measured by optical density at a wavelength of 475 nm and a reference wavelength of 650 nm using a spectrophotometer (Infinite M200, Tecan).

### 2.8. Verification of Mesenchymal Stromal Cell Classification of Sprayed Cells

The isolation protocol for ASCs was verified as defined by Dominici et al. [18]. To ensure that mesenchymal stromal cell classification regarding specific surface marker expression is preserved after spraying, non-sprayed and sprayed cells in passage 2 were characterized analogously by flow cytometry. Briefly, ASCs were cultured for 7 days, trypsinized, fixed in 4% paraformaldehyde (PFA, Carl Roth, Karlsruhe, Germany) buffered with PBS for 10 min at RT, resuspended in 2% FCS-PBS and stained with the respective antibodies for CD90, CD73 and CD105, CD45, CD34, CD11b, CD79α, and HLA-DR surface molecules (all mouse anti-human, BD Biosciences), analyzed by flow cytometry with a FACS Canto II and evaluated with FlowJo v10 (both Becton Dickinson).

### 2.9. Evaluation of Tri-Lineage Cell Differentiation Capacity of Sprayed Cells

To ensure that stromal cell multipotent differentiation capacity of ASCs is preserved after spraying, sprayed cells were differentiated into osteogenic, adipogenic, and chondrogenic cells using the StemPro Osteogenesis and Adipogenesis Differentiation Kits (both ThermoFisher) and the MesenCult-ACF Chondrogenic Differentiation Kit with L-Glutamine (both STEMCELL Technologies), respectively. Differentiation kits were used according to the manufacturer’s instructions. For all kits, non-sprayed cells and sprayed cells incubated with full Mesenpan medium served as controls. Differentiation capacity was evaluated using histological methods.

#### 2.9.1. Osteogenesis

ASCs were seeded in 12-well plates at 5 × 10^3^ cells/cm^2^ with a working volume of 1 mL and incubated with full Mesenpan medium at 37 °C and 5% CO_2_. After 24 h, medium was replaced with differentiation medium (basal medium with supplements and 5 µg/mL Gentamicin) and medium was exchanged three times a week. After 24 days, cells were rinsed with PBS and fixed with 4% PFA-PBS for 30 min at RT. After fixation, cells were rinsed with distilled water and stained with 2% Alizarin Red S for 3 min and again rinsed with water.

#### 2.9.2. Adipogenesis

ASCs were seeded in 12-well plates at a density of 10^4^ cells/cm^2^ with a working volume of 1 mL and incubated with full Mesenpan medium at 37 °C and 5% CO_2_. After 24 h, medium was replaced with differentiation medium (basal medium with supplements and 5 µg/mL Gentamicin) and medium was exchanged three times a week. After 10 days, cells were fixed with 4% PFA-PBS for 10 min at RT and rinsed with PBS. For staining, Oil Red O (Sigma-Aldrich) was dissolved in isopropanol with a concentration of 0.5 g/100 mL and diluted with distilled water at a ratio of 3:2 and let sit for 10 min. The solution was filtered through cellulose. The samples were stained for 15 min at RT and washed with distilled water.

#### 2.9.3. Chondrogenesis

Chondrogenesis of ASCs was performed in pellet culture within centrifugation tubes. ASCs suspended in MesenCult-ACF chondrogenic differentiation medium (including 2 mM L-Glutamine and 1% ABM) or full Mesenpan medium (control) at a concentration of 10^6^ cells/mL were transferred to a centrifugation tube at a total volume of 0.5 mL and centrifuged for 10 min at 300× *g*. Cells were cultured at 37 °C and 5% CO_2_ for 21 days with an only lightly screwed cap to ensure gas exchange. After 24 h, 0.5 mL of the respective medium was added to the tube and three times a week 0.5 mL of the respective medium was exchanged. After 25 days, cell pellets were fixed with 4% PFA-PBS for 30 min at RT, rinsed with PBS, and dehydrated in an ascending isopropanol series (VWR) with subsequent xylene (Fischar, Saarbrücken, Germany) incubation before embedding in paraffin (Tissue-Tek, Sakura, Umkirch, Germany). Paraffin sections (5 µm) were dewaxed in xylene, hydrolyzed in a descending ethanol series, stained with Alcian Blue (Waldeck, Münster, Germay) and Nuclear Fast Red (Roth), and counterstained with hematoxylin (Sigma-Aldrich).

One cell pellet for gas-to-liquid ratio 10% cultured in full-growth medium was lost during the embedding process due to disintegration. As a less compact cell pellet indicates an undifferentiated ASC pellet, this was not considered further.

### 2.10. Statistical Analysis

All cell experiments were carried out with three independent biological donors. Results are presented as means ± standard deviations (SD). Data were analyzed using Excel 2016 (Microsoft) and Prism 9 (Version 9.4.0, Graphpad Software). All response variables were tested for normality using the Shapiro–Wilk statistic. One-way ANOVA with Dunnett’s multiple comparisons test was carried out for apoptosis and necrosis assays. Two-way ANOVA with Dunnett’s and Tukey’s multiple comparisons test was used for metabolic activity assay and survival rate, respectively. The statistical analysis was conducted in an explorative manner, and a *p*-value below 0.05 was considered to be statistically significant (labeled *).

## 3. Results

### 3.1. Spray Visualization and Droplet Size Distribution

With high-speed spray imaging, the sprays were characterized with regard to their droplet size and distribution (Figure 4). The droplet sizes D_10_ (arithmetic mean diameter) as well as the share of droplet sizes are given for the gas-to-liquid ratios from 2.5% to 10% and mixing tubes A, B, and C in Figure 5a.

D_10_ decreases with increasing gas-to-liquid ratio for all mixing tubes. For gas-to-liquid ratios of 5% up to 10%, this decrease is continuous for each mixing tube. Here, mixing tube B shows the smallest D_10_ compared to mixing tubes A and C. In detail, D_10_ ranges from 33.5 µm (mixing tube B) and 24.4 µm (mixing tube A) for a gas-to-liquid ratio of 2.5% and 18.9 µm (mixing tube C) and 15.4 µm (mixing tube B) for the gas-to-liquid ratio 10%. 

The histograms (Figure 5b–d) show the share of droplet sizes for the analyzed sprays within a range of 6 and 500 µm and a 2-µm bin. Independent of mixing tube or gas-to-liquid ratio, all histograms exhibit an accumulation of droplets with a diameter range of 6 to 20 µm as well as droplets with a diameter of above 100 µm that are summarized as one value. For the latter, a maximum of 7.4% was observed for mixing tube B, and a gas-to-liquid ratio of 2.5%. For all spray conditions, droplets with diameters of about 40 to 100 µm show only a low frequency. With an increasing gas-to-liquid ratio, a slight left shift of the histograms is observed. Thus, there are more small droplets for higher gas-to-liquid ratios. Therefore, the absolute numbers of analyzed droplets vary from 14,543 for a gas-to-liquid ratio of 2.5% to 57,801 for a gas-to-liquid ratio of 10% on average.

With 75.9% of droplets with a diameter of ≤16 µm, mixing tube B with a gas-to-liquid ratio of 10% has the largest proportion of small droplets. Mixing tube A and mixing tube C show smaller fractions of small droplets with 69.5% and 64.1%, respectively.

### 3.2. Cell Survival

To show the general suitability of the effervescent atomizer for cell spray technology and to identify the spray parameters with the highest survival rates, spray parameters (gas-to-liquid ratios and mixing tubes) were varied and cell survival of sprayed ASCs was evaluated by flow cytometry analysis. Non-sprayed cells served as controls. Figure 6 shows the survival rates of sprayed ASCs (*n* = 3) for the different mixing tubes (A, B, and C) and gas-to-liquid ratios of 2.5 to 10%, and the survival rate of the non-sprayed control (as the measurements for the mixing tubes were carried out at different time points, there is a separate control for each mixing tube). In general, the survival rate is continuously decreasing with the increasing gas-to-liquid ratio for all mixing tubes. Compared to mixing tube A and mixing tube C, mixing tube B features higher survival rates for all gas-to-liquid ratios, though without statistical significance. Compared to the respective control, for mixing tube B, there was no significantly reduced survival for gas-to-liquid-ratios 2.5% and 5%. For A, it was reduced for all conditions, and for C from 5%. Because of this and the consistently higher survival rates of mixing tube B compared to mixing tube A and mixing tube C, the following experiments were carried out with mixing tube B.

### 3.3. Cell Behavior

The cytocompatibility of the atomization process was evaluated by assessing the cell health of sprayed ASCs based on metabolic activity, apoptosis, and necrosis (Figure 7). Cell proliferation of sprayed ASCs was evaluated over a culture period of six days after atomization by the conversion of XTT to formazan in the presence of metabolic activity. The metabolic activity is normalized to non-sprayed ASCs cultured in full-growth medium on the first measurement day (48 h post spraying). Incubation with a cytotoxic control (latex) revealed no metabolic activity of the cells.

On day two, the metabolic activity of the sprayed cells is significantly decreased for all spray conditions (gas-to-liquid ratio 2.5–10%), with a continuous decrease of activity over the increasing gas-to-liquid ratios. Over the culture period, the differences to the non-sprayed control decrease. Within the different spray conditions (gas-to-liquid ratio 2.5–10%), a decrease in metabolic activity is observable with increasing gas-to-liquid ratio for all three measurement days. However, the differences between different gas-to-liquid ratios decrease with an increased culture period.

The assays for apoptosis and necrosis detect substances released to the culture medium by apoptotic and necrotic cells, respectively. Hence, assays were performed after 24 h and thus before the need for a medium exchange. Signals are normalized to apoptosis- and necrosis-induced controls.

Compared to the apoptosis-induced control, non-sprayed and sprayed cells, independently of the spraying parameters, show only low caspase 3/7 activity between 11.8 ± 6.0%, and 16.6 ± 3.0% for gas-to-liquid ratio 5 and 10%, respectively. Statistical analysis shows no significant differences between non-sprayed and sprayed cells. In contrast, necrosis of sprayed cells increased continuously with increasing gas-to-liquid ratios from 22.0 ± 6.2% up to 55.8 ± 6.4% for gas-to-liquid ratios of 2.5 and 10%, respectively. The non-sprayed control shows the lowest necrosis level of 9.6 ± 1.8%. Except for the gas-to-liquid ratio of 2.5%, statistical analysis shows significant differences between the non-sprayed control and the sprayed cells. 

### 3.4. Verification of Mesenchymal Stromal Cell Classification of Sprayed Cells

The flow cytometric analysis of non-sprayed and sprayed (gas-to-liquid ratios 2.5% and 10%) cells with regard to mesenchymal stromal cell-characteristic surface markers is given in Figure 8. In all cases, ASCs were positive for CD90, CD73, and CD105 with ≥98.5%. The markers CD45, CD34, CD11b, CD79α, and HLA-DR surface molecules were negative (≤0.1%). The exact values for all cell lines and representative dot plots of cytometric analysis are given in the Appendix A.

### 3.5. Verification of Tri-Lineage Differentiation Potential

Medium-induced tri-lineage differentiation of non-sprayed and sprayed (gas-to-liquid ratios 2.5% and 10%) into adipocytes, osteocytes, and chondrocytes was performed according to the manufacturers’ instructions. The differentiation was evaluated using histological methods and bright-field imaging (Figure 9).

All cells cultured in their respective differentiation medium show positive staining. For adipocytes, vacuoles filled with lipids were stained by Oil Red O. The differentiation into osteocytes was proven by the staining of calcium by Alizarin Red S. In chondrocytes, proteoglycans show blue staining by Alcian Blue. The intensity of staining differs between samples and was most intense for highly confluent areas. For all donors, the non-sprayed ASCs differentiated into adipocytes showed less intense staining in comparison to the sprayed samples. Some cell layers of the osteogenic differentiation detached from the well bottom as previously observed for over-confluent samples after extended culture periods. This was not considered for the evaluation as all remaining samples showed differentiation into osteocytes. For all controls cultured in the full-growth medium, the respective staining was negative (Appendix A).

## 4. Discussion

In this study, we used the effervescent atomization principle for cell spraying for the first time. It was shown that this technique enables cell application under sterile and reproducible conditions for different parameters. The cell behavior after aerosolization was highly dependent on the testing conditions.

The three different mixing tubes varied in terms of the number of holes for gas-air mixing and hole diameters. The droplet sizes were mainly dependent on the gas-to-liquid ratio, with decreasing droplet sizes for increasing ratios. For the arithmetic mean diameter, values were in a similar range for all mixing tubes (especially A and C) and no clear trend was visible between the mixing tubes. For mixing tube B, the arithmetic mean diameter was highest for a gas-to-liquid ratio of 2.5%, but lowest for the higher gas-to-liquid ratios. Due to the chosen design principles based on Jedelsky et al. [15], mixing tube C was expected to express the lowest D_10_. Thus, the geometrical dimensions of this atomizer are much smaller than those developed in Jedelsky et al. Other effects, such as the boundary layer in the vicinity of the wall could influence the atomization behavior.

Note that the number distribution of the droplets can only be used to a limited extent to evaluate or compare the cell survival rate. As already shown by Gürzing et al., the cell survival rate does not correlate directly with the arithmetic mean droplet diameter, but with the mean droplet volume of the spray [11]. However, a measurement of the droplet volume or the droplet volume distribution is only appropriate if the measurement system guarantees a high counting efficiency for large liquid structures. However, this is not given in the present case of effervescent atomizers, which are characterized by a pulsating outlet flow. At certain times, this pulsating flow leads to comparatively large liquid structures that cannot be detected by the measuring system (partly due to blurred edges, partly due to incomplete imaging in the field of view). For this reason, this work refrains from calculating volume-averaged droplet diameters, such as Sauter diameter or mean volume diameter of the spray.

In the histograms, it can be observed that the majority of the droplets was smaller than 20 µm for all mixing tubes. Nevertheless, there was also a considerable fraction of droplets with sizes of 100 µm and higher. ASCs in suspension in a low passage are described to have a diameter of approximately 26 µm [19]. With survival rates of more than 90% for the lowest gas-to-liquid ratio, the comparably small droplets seem to not harm the general suitability of these spray characteristics for ACS aerosolization. 

Nevertheless, the decrease in droplet sizes which is observed for the arithmetic mean diameter is reflected by decreasing cell survival rates for increasing gas-to-liquid ratios. This also indicates a further decline in cell survival rate for gas-to-liquid ratios of more than 10%. Thus, successful cell aerosolization is not expected for higher ratios because stresses on the cells increase. This is associated with the generation of continuously smaller droplets until the droplets cannot contain whole cells anymore. In a previous study by the authors, A549 cells were sprayed using a coaxial atomizer. Survival rates of more than 90% and 85% for gas-to-liquid ratios of 2.95% and 4.84% were reported, respectively [14]. Thus, a decreasing tendency of the survival rate with increasing gas-to-liquid ratios was observed for the coaxial as well as for the effervescent atomizer.

In general, it has to be noticed that results from different studies are difficult to compare, as the outcome depends on the cells, substrate, distance to substrate, medium for aerosolization, and obviously atomizer design. In this study, we were able to prove the general suitability of the effervescent atomizing principle with the advantage over other atomizer systems of reduced air, which can prevent quick evaporation and draining of the sprayed liquid.

Overall, the survival rates shown here, especially for a gas-to-liquid ratio of 2.5% are well within a range considered suitable for the clinical application of cell therapy approaches [20]. In our previous study, the atomization with both tested atomizers lead to comparable survival rates of more than 90% under optimized conditions [8]. In a study by Aguado et al., it was shown that the ejection of ASCs through a 28-Gauge needle already reduces the cell survival of ASCs to less than 80% [21]. Thus, not only shear and elongation forces during droplet formation but also during tube flow can decrease cell survival [22]. With the orifice diameter of 0.5 mm used for all mixing tubes here, the influence of this factor can be regarded as limited.

Different from our previous study, the metabolic activity decreased significantly on day two for all spray conditions [8]. This effect was only visible for the two higher gas-to-liquid ratios on day four and no effect was observed anymore after six days. As the reduction in metabolic activity is higher for the higher gas-to-liquid ratios, this could be a consequence of the decreased survival rates directly after spraying or by a reduction in the total number of cells during the processing.

Necrosis levels increased with increasing gas-to-liquid ratios. For gas-to-liquid ratios higher than 2.5%, the difference to the non-sprayed control was significant. Necrosis is characterized by an increased plasma membrane permeability and the release of intracellular content [23]. This effect could be caused by cell (membrane) disintegration initiated by the hydrodynamic forces as well as shear and elongation stresses acting on the cells. This correlation of increasing necrosis levels with increasing gas-to-liquid ratios is supported by the same tendency for the survival rate (decrease for increasing gas-to-liquid ratios). Apoptosis levels were not influenced by the spray process. Thus, the forces acting on the cells during application with the effervescent atomizer do not induce programmed cell death. Thus, the stresses acting on the cells result rather in direct cell disintegration instead of inducing apoptosis. Similarly, Kim et al. have shown that apoptosis rates were not increased on day three for mesenchymal stromal cells processed with a Teleflex atomizer [9]. In our previous study, we observed significantly increased apoptosis rates for mesenchymal stromal cells, which were aerosolized with a pressure-based atomizer, whereas the coaxial atomizer did not induce apoptosis [8].

As one of the minimal criteria defined by Dominici et al., the expression of specific cell surface markers needs to be fulfilled [18]. Here, we were able to show that the expression pattern of ASCs is not changed by cell atomization. Tri-lineage differentiation into adipocytes, osteocytes, and chondrocytes was successful for all conditions. Thus, atomization has no influence on the stem-ness of ASCs. In addition, no differentiation occurred in the normal culture medium just by induction of the spraying. In previous studies, it has been shown that differentiation of mesenchymal stem/stromal cells can be induced by mechanical stimulation [24]. Here, the time or extent of stresses seem to not influence cell differentiation into osteocytes or chondrocytes and does not harm their differentiation potential. For adipogenesis though, an increased histological staining was visible.

In an asthma model, aerosolized ASCs were proven to reduce airway inflammation by means of tissue thickness, granulocyte infiltration, and levels of pro-inflammatory genes [25]. Possible paracrine effects after in vivo/in situ application with the effervescent device still have to be evaluated. Here, immunomodulatory effects in vitro and in vivo are crucial to evaluate the potential of this aerosolization principle.

The setup as shown in this study is only suitable for cell application onto exposed surfaces. For our experiments, having a fixed distance to the substrate was very important such that the equipment seems to be quite complex. For clinical use, this can be easily adapted by different measures: Currently, the pressurized air is fed from two sides; this can be reduced to one air connection with a longer tube for hand-held usage. With other manufacturing methods, the mixing tube and nozzle can be produced as one part, thereby reducing the outer dimensions to several millimeters only. In general, the spraying principle would allow miniaturization of the housing and connectors even for endoscopic use.

In this study, the effervescent atomization principle was evaluated and proven suitable for cell application with reduced amounts of supplied air compared to conventional atomizers. To the authors’ knowledge, this was the first time an effervescent atomizer was used for cell processing. Three atomizer designs were manufactured and evaluated with gas-to-liquid ratios between 2.5 and 10% regarding the spray characteristics and the influence of the processing on ASCs. Droplet diameters decreased for increasing gas-to-liquid ratios. After aerosolization, the survival rate of ASCs was >90% of the control for the lowest gas-to-liquid ratio. For higher ratios, the cell survival decreased to 51.1%. Metabolic activity was significantly reduced in the short term but normalized until day seven. The apoptosis rates were not influenced by aerosolization, whereas high gas-to-liquid ratios caused increased necrosis levels. Tri-lineage differentiation potential into adipocytes, chondrocytes, and osteoblasts was not negatively influenced by aerosolization.

## Figures and Tables

**Figure 1 pharmaceutics-14-02421-f001:**
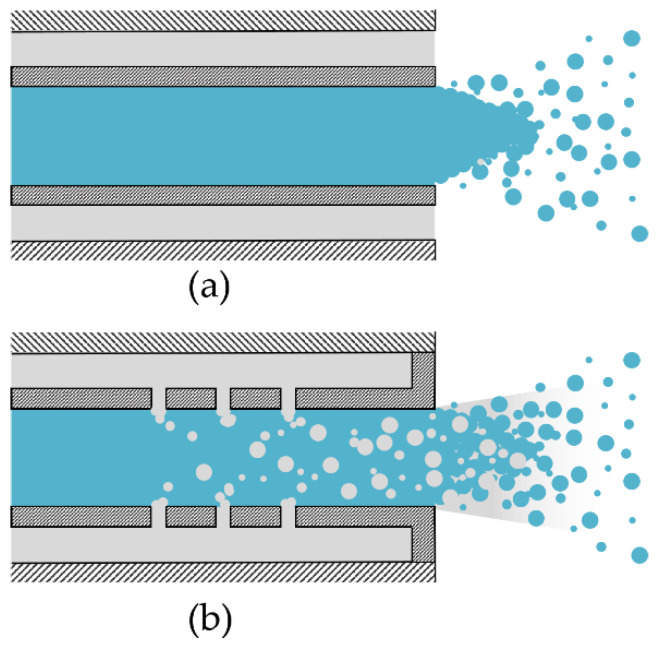
Atomization Principles. (**a**): Coaxial atomization: air and liquid mix at the outlet of the liquid tube (external mixing). (**b**): Effervescent atomization: air is bubbled into the liquid before ejection (internal mixing).

**Figure 2 pharmaceutics-14-02421-f002:**
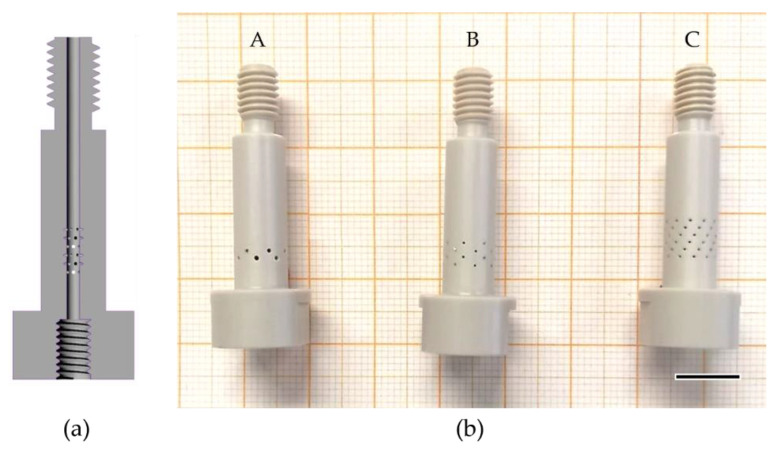
(**a**) Schematic of custom-made mixing tubes. (**b**) Three different designs: Mixing tubes A, B, and C. Scale bar: 10 mm.

**Figure 3 pharmaceutics-14-02421-f003:**
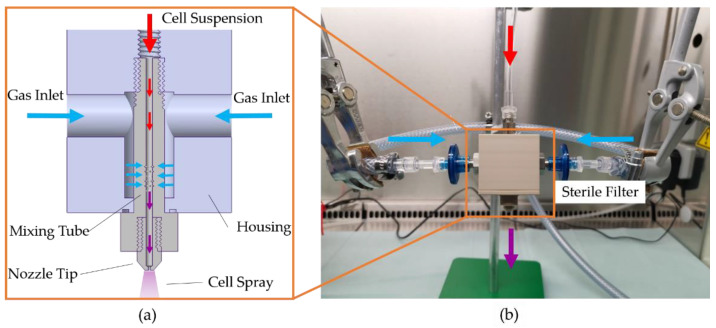
(**a**) Drawing of the effervescent atomizer including housing, mixing tube, and nozzle tip. (**b**) Experimental setup for cell spraying within the laminar flow hood.

**Figure 4 pharmaceutics-14-02421-f004:**
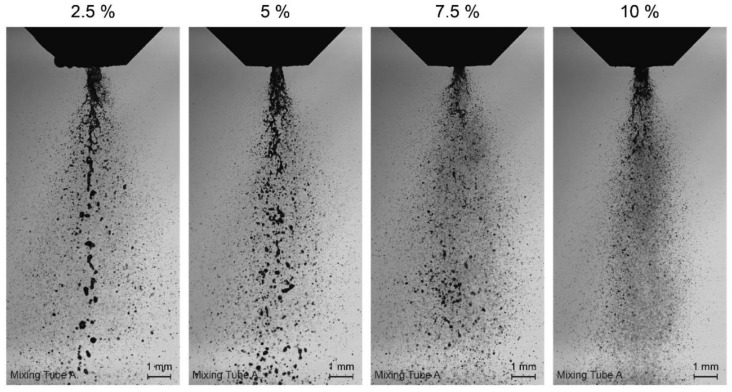
High-speed visualization of droplet size distribution for gas-to-liquid ratios 2.5 to 10%. Images for mixing tube A are shown as example for the influence of the gas-to-liquid ratio.

**Figure 5 pharmaceutics-14-02421-f005:**
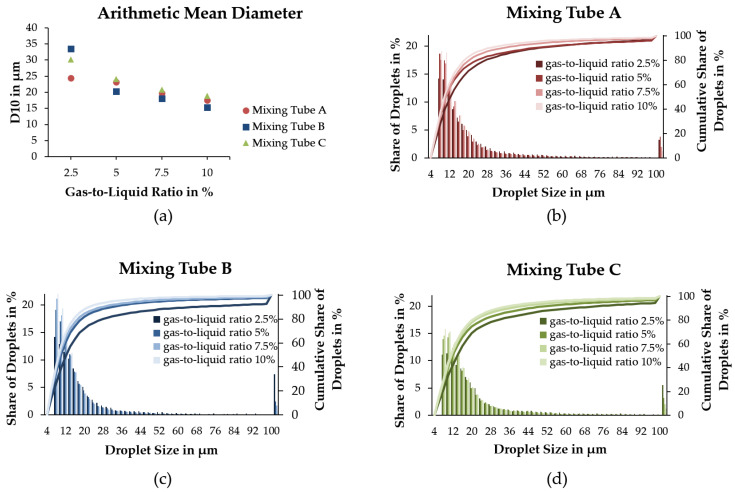
Mean droplet diameter of spray given for the arithmetic mean D_10_ (**a**). Note the different scaling. The share of droplets and cumulative share of droplets in dependence of droplet diameter for mixing tubes A, B, and C (**b**–**d**, respectively). The values given for 100 µm include all droplets between 100 to 500 µm.

**Figure 6 pharmaceutics-14-02421-f006:**
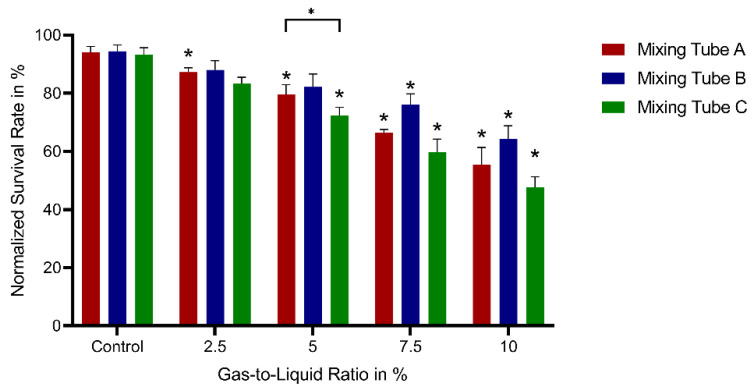
Normalized survival rates of sprayed ASCs (*n* = 3) for different mixing tubes (A, B, C) and gas-to-liquid ratios. Significant differences with *p* < 0.05 were considered as significant and are marked with *. Asterisks above a bar refer to significant differences compared to the respective control. Values were compared for each gas-to-liquid ratio.

**Figure 7 pharmaceutics-14-02421-f007:**
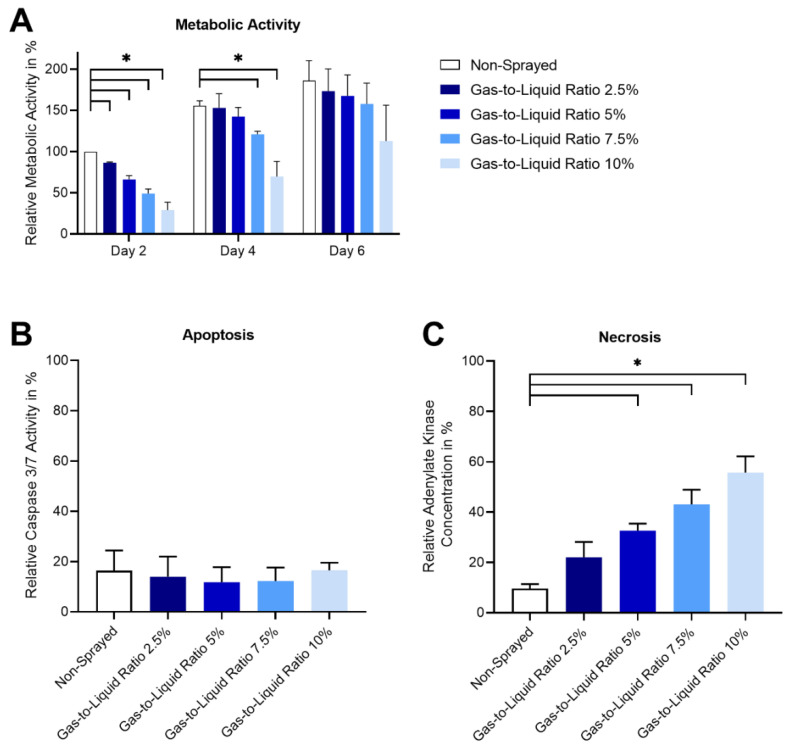
Cell proliferation, apoptosis, and necrosis after atomization of ASCs (*n* = 3) with mixing tube B and different gas-to-liquid ratios. Cell proliferation was evaluated by metabolic activity of cells and normalized to non-sprayed cells on the first day of measurement (**A**). Apoptosis was analyzed after 24 h by caspase 3/7 activity and normalized to a staurosporine-incubated control (**B**). Necrosis was evaluated after 24 h by adenylate kinase release and normalized to a 100%-lysis control (**C**). Significant differences to the on-sprayed control are marked with * (*p* < 0.5).

**Figure 8 pharmaceutics-14-02421-f008:**
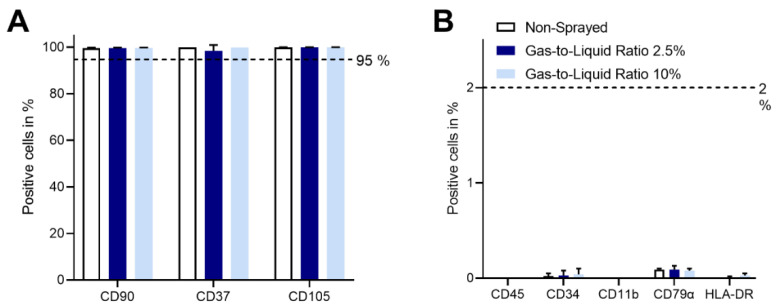
Percentage of positive ASCs (*n* = 3) cells stained for eight mesenchymal stromal cell-characteristic markers of non-sprayed and sprayed with gas-to-liquid ratio 2.5% and 10% after 7 days of culture. Flow cytometric analysis shows that all experimental conditions fulfill the minimal criterion defined by Dominici et al. [18] regarding the surface marker expression; CD90, CD37, and CD105 are expressed in more than 95% (**A**), CD45, CD34, CD11b, CD79 α, and HLA-DR in less than 2% of cells (**B**).

**Figure 9 pharmaceutics-14-02421-f009:**
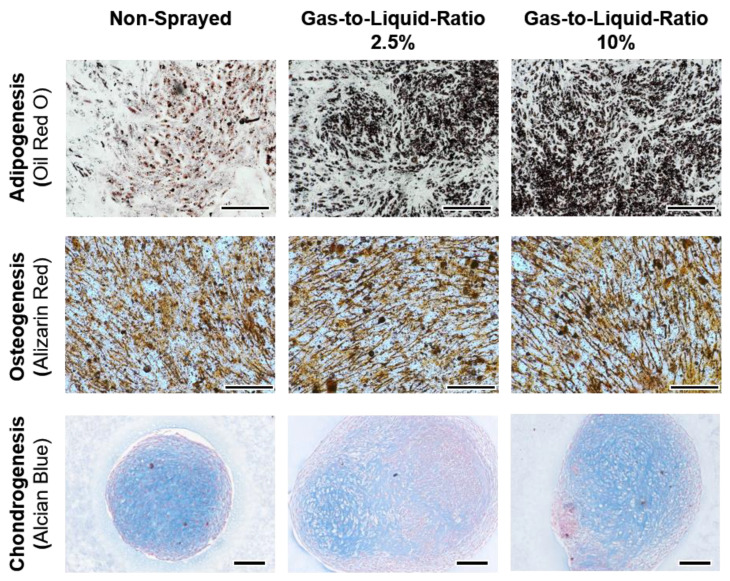
Non-sprayed and sprayed (gas-to-liquid ratio 2.5% and 10%) ASCs after tri-lineage differentiation into adipocytes (top row), osteocytes (middle row, scale bars: 500 µm), and chondrocytes (bottom row, scale bar: 200 µm).

**Table 1 pharmaceutics-14-02421-t001:** Design specifications of custom-made effervescent atomizer with three different mixing tubes.

Mixing Tube	A	B	C
Diameter of Exit Orifice in mm	0.5	0.5	0.5
Diameter of Mixing Tube in mm	1.5	1.5	1.5
No. Drilling Circles	2	4	6
No. Holes per Drilling Circle	8	8	12
Total No. of Holes	16	32	72
Hole Diameter in mm	0.55	0.40	0.3
Total Area of Holes in mm^2^	3.80	4.02	5.09

**Table 2 pharmaceutics-14-02421-t002:** Overview of investigated spray parameter sets.

Gas-to-Liquid Ratio in %	Flow Rate Fluid in mL/min	Flow Rate Air in L/min	Excess Air Pressure in Bar
2.5	24	0.6	1.5
5	24	1.2	1.5
7.5	24	1.8	1.5
10	24	2.4	2

## Data Availability

The data presented in this study are available on request from the corresponding author.

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
