# Peer review of "Effervescent Atomizer as Novel Cell Spray Technology to Decrease the Gas-to-Liquid Ratio"

_pharmaceutics, 2022, doi:10.3390/pharmaceutics14112421_

Round 1

Reviewer 1 Report

Overall

The study investigates a modification to the cell spraying process for therapeutic cell delivery. I believe the study has novelty in the testing of a new spray design, which builds on previous studies. There is important data here for researchers investigating cell spraying and it could have wider interest to those studying cell therapies. The whole paper is also well written and in general well presented. I believe the work can be published, although I have a few comments and questions which it would be good if the authors can address prior to publication.

Comments:

1.       I think the intro should have a little more information on why cell spraying has good potential as a cell delivery approach. This is especially important given the paper is focussed on a way to optimise cell spraying for therapeutic delivery.

2.       Can the authors clarify what “massive inflation” means? (Introduction, line 74)

3.       This may be my ignorance, but in figure 6 can the D10 be displayed with error bars and statistically analysed from one condition to the next?

4.       Can the authors comment on how reproducible the droplet sizes that are produced are? For example, was the experiment performed on different days to test whether the D10 and associated histograms differed from run to run? Also, how many frames were analysed to produce the histogram?

5.       This may be personal opinion and if the editor is happy then that is OK but I don’t like the way the asterisk is used to show statistical significance in the graphs. In fig 5, it is difficult to see which bar is significantly different to which other bar. For the 7.5%, are they all significantly different from each other? Or are they all different from 2.5%? Perhaps greater detail could be provided in the legend to also clarify this.

6.       There is a case in figure 5 to use a 2 way ANOVA as there are two variables (mixing tube and gas-liquid ratio). Did the authors try this analysis?

7.       Also, why do you need to normalise the cell survival? Could you just add non-sprayed to the graph? I realise you have provided the non-sprayed viability values in the text, but then I have to do additional maths to work out the sprayed viability. If you have non-sprayed in there as well, you can more strongly say that the viability is not affected by spraying (if it isn’t).

8.       For the cell health analyses (live staining, apoptosis and necrosis), it would be helpful to show some pictures if possible to indicate what the staining looks like. This is important given the gating must need to be set against a known fluorescence event and give us an impression of the health.

9.       The MSC  differentiation figure should be figure 9.

10.   You say the “amount of differentiation was evaluated using histological methods”. How d you assess “amount” in this context? Also, were these experiments also repeated three times?

11. The spray equipment seems quite complex (fig 3b). Can the authors add some lines to the discussion on how this might be adapted for clinical use?

Author Response

We sincerely thank the reviewers for the interesting and helpful comments; we appreciate the time and effort that the reviewers have put into this feedback. Upon reviewing the comments, we found there were several areas in the manuscript that could be improved upon. We have responded to and addressed all the remarks made by the reviewers below and in the revised manuscript; changes are highlighted in yellow.

Reviewer 2 Report

Current article provides sound hypothesis and scientific results. Therefore, it can be published as it is. 

Author Response

The authors would like to thank the reviewer for the positive evaluation.

Reviewer 3 Report

The manuscript by Thiebes et al. reported a novel custom-made atomizer based on the effervescent principle. Different designs were evaluated regarding spray characteristics and influence on human adipose-derived mesenchymal stromal cells. The survival, differentiation and cell surfaces marker were analyzed before and after the spraying process. I made considerations so that the authors can think about it and seek to improve the study.

1. The author should clearly state the disadvantages of other aerosolization methods, such as a coaxial atomizer. It would be better to compare the droplet size and distribution as well as survival rates of these two aerosolization methods.  

2. If such homogenized atomization is beneficial in the clinical application at the expense of reduced survival? There was significant increase of necrosis after cell spraying by effervescent atomizer, does that mean a lot of cells have been lost during the atomization process? A comparison of the total cell number before and after spray should be included..

3. A comparison of the cell surface markers such as CD90, CD105 etc. seems to be unimportant. A spraying process would interfere with the biological functions instead of the purity of sprayed cells. In terms of the application of reducing airway inflammation, the immunosuppressant function of MSCs should be evaluated.

4. In Fig. 8, the adipogenesis capacity of MSCs showed obvious difference in gas-to-liquid-ration 2.5% group as compared to the control, indicating the MSCs function has been modified by the spraying process.

5. Line 119  the flow rate of the spray fluid was set to 22.4ml/min. Could the author explain why choose such a flow rate ?

6. The principles of designing the custom-made effervescent atomizer should be explained. The number of the drilling holes and the diameter of holes are both changed in the mixing tubes A, B and C.

Author Response

(The authors gave the same response as above.)

Round 2

Reviewer 3 Report

no further comments